# Clinical Characterization and Founder Effect Analysis in Chinese Patients with Phospholipase A2-Associated Neurodegeneration

**DOI:** 10.3390/brainsci12050517

**Published:** 2022-04-19

**Authors:** Hao-Ling Cheng, Yi-Jun Chen, Yan-Yan Xue, Zhi-Ying Wu, Hong-Fu Li, Ning Wang

**Affiliations:** 1Department of Neurology, Institute of Neurology, The First Affiliated Hospital of Fujian Medical University, Fuzhou 350005, China; haolingcheng@fjmu.edu.cn (H.-L.C.); yijun_chen2015@163.com (Y.-J.C.); 2Research Center of Neurology, Key Laboratory of Medical Neurobiology of Zhejiang Province, Department of Neurology, Second Affiliated Hospital of Zhejiang Province, Zhejiang University School of Medicine, Hangzhou 310000, China; yanyanxue@zju.edu.cn (Y.-Y.X.); zhiyingwu@zju.edu.cn (Z.-Y.W.); 3Fujian Key Laboratory of Molecular Neurology, Fujian Medical University, Fuzhou 350005, China

**Keywords:** phospholipase A2-associated neurodegeneration, *PLA2G6*, clinical features, founder effect, Chinese

## Abstract

*PLA2G6*-associated neurodegeneration (PLAN) is a rare autosomal recessive disorder caused by *PLA2G6* mutations. This study aimed to investigate the clinical characteristics and mutation spectrum of PLAN and to investigate the founder effects in Chinese PLAN patients. Six Chinese PLAN families were clinically examined in detail and whole-exome sequencing was performed in the probands. Haplotype analysis was performed in five families with the *PLA2G6* c.991G > T mutation using 23 single nucleotide polymorphism markers. Furthermore, all previously reported *PLA2G6* mutations and patients in China were reviewed to summarize the genetic and clinical features of PLAN. Interestingly, we found that one patient had hereditary spastic paraplegia and showed various atypical clinical characteristics of PLAN, and five patients had a phenotype of parkinsonism. All probands were compound heterozygotes for *PLA2G6* variants, including four novel pathogenic/likely pathogenic mutations (c.967G > A, c.1450G > T, c.1631T > C, and c.1915delG) and five known pathogenic mutations. Haplotype analyses revealed that patients carrying *PLA2G6* c.991G > T mutations shared a haplotype of 717 kb. The frequencies of psychiatric features, cognitive decline, and myoclonus in Chinese patients with *PLA2G6*-related parkinsonism were significantly different from those in European patients. Thus, our study expands the clinical and genetic spectrum of PLAN and provides an insightful view of the founder effect to better diagnose and understand the disease.

## 1. Introduction

Phospholipase A2 group VI (*PLA2G6*)-associated neurodegeneration (PLAN; MIM*603604) is a heterogeneous group of disorders caused by biallelic mutations in *PLA2G6* [1,2]. It is estimated that the disease prevalence rate is approximately 1/1,000,000 [3]. Based on the age of onset and clinical manifestations, PLAN can be divided into three major subtypes: infantile neuroaxonal dystrophy (INAD), atypical neuroaxonal dystrophy (ANAD) and *PLA2G6*-related parkinsonism [1]. INAD and ANAD patients present with childhood-onset progressive psychomotor deterioration, axial dystonia, cognitive impairment, ataxia, spastic quadriparesis, and occasional optic atrophy [1,4]. *PLA2G6*-related parkinsonism includes autosomal recessive early-onset parkinsonism (AREP, known as PARK14) and adult-onset dystonia-parkinsonism (DP), and these patients may present with parkinsonism, dystonia, cognitive regression, and gait instability [1,5]. PLAN often shows cerebellar atrophy and evidence of iron accumulation on brain magnetic resonance imaging (MRI) [1]. Moreover, new cases are increasingly reported and the phenotype is expanding, including hereditary spastic paraplegia (HSP), and ataxia with or without brain iron accumulation [6,7,8]. Given the extremely heterogeneous clinical features, the diagnosis of PLAN can be confirmed by detecting mutations in the *PLA2G6* gene.

The *PLA2G6* gene is located on chromosome 22q13.11, consisting of 17 exons, encoding an 85/88 kDa protein and spanning more than 69 kb [9]. Since the identification of *PLA2G6* mutations among two Israeli INAD families in 2006 [10], over 200 different mutations in PLAN patients have been identified in *PLA2G6* according to the Human Gene Mutations Database (HGMD^®^ Professional 2021.4, http://www.hgmd.org, accessed on 1 March 2022). The mutation spectrum varies according to the country of origin. For example, a recurrent *PLA2G6* mutation c.2070_2072delTGT (p.V691del) has been suggested as a founder mutation in the North African population [11], but this mutation is rare in Chinese population. However, the c.991G > T variant in *PLA2G6* was almost exclusively found in Chinese patients [6,12], but it remains to be established whether c.991G > T is a potential founder or a hotspot mutation in the Chinese population. Moreover, PLAN patients of Chinese ancestry are rarely documented (less than 70) (Appendix A) [6], and genetic analysis and clinical characterization of these patients are limited.

Thus, this study aimed to investigate the genetic and clinical features of six unclear Chinese patients with PLAN and the founder effect of the *PLA2G6* most frequent mutation in the Chinese population, and to summarize the genotypes and phenotypes of all Chinese patients with PLAN. Our results contribute to expanding the knowledge of genetics and clinical in PLAN and consequently could be useful in guiding clinical decision-making and improving clinical diagnosis.

## 2. Materials and Methods

### 2.1. Participants

A total of 6 novel clinically suspected PLAN pedigrees of Chinese ancestry, including 6 patients and 14 family members, were consecutively collected between August 2017 and August 2021 from the Department of Neurology in the Second Affiliated Hospital of Zhejiang University School of Medicine. All probands had no blood relationship and no parental consanguinity was noted in any family. Each patient was given detailed clinical evaluations and neurological examinations by at least two senior neurologists. Written informed consent was signed by each participant or their legal guardian. The study was approved by the Ethics Committee of the Second Affiliated Hospital of Zhejiang University School of Medicine. For the haplotype studies, we enrolled a PLAN family with the *PLA2G6* c.991G > T mutation, which was previously described in our center [6].

### 2.2. Molecular Genetics

Genomic DNA was extracted from peripheral blood samples of probands and the available family members using a QIAamp DNA Blood Minikit (QIAGEN, Hilden, Germany). Whole-exome sequencing (WES) was performed in the index patient, and subsequent variant calling, alignment, and annotation were performed, as described previously [13]. Sanger sequencing was used to verify the potential variants and to determine the co-segregation of the pedigree. *PLA2G6* variants are named according to the GenBank: NM_003560 reference transcript. The VarSome and InterVar tools, which are based on American College of Medical Genetics and Genomics (ACMG) [14], were used to interpret variants [15,16].

### 2.3. Haplotype Analysis

Single nucleotide polymorphisms (SNPs) within the 2.3-Mbp region (Chr22_37407502-39701697) surrounding the position of the most frequent mutation (c.991G > T) were obtained from the data of the southern Han Chinese (CHS) population based on the 1000 Genomes, human assembly GRCh37. The tagger function of Haploview 4.2 software was used to analyze the tag SNPs [17]. The 23 most representative tag SNPs (11 sites upstream and 12 sites downstream) tagging other common SNPs (minor allele frequency > 0.3) in these loci with r² > 0.8, and were selected at approximately 100,000 bp intervals. Haplotype analysis was performed using the Sanger sequencing method.

### 2.4. Literature Review

We systematically searched PubMed for Chinese patients with genetically confirmed PLAN published before March 2022. The search strategy was “*PLA2G6*” AND “Chinese” regardless of the language restriction. Additional references to related articles were identified and reviewed. Only PLAN cases caused by biallelic *PLA2G6* mutations and reported individual information were included. The report of Magrinelli et al., in January 2022 is the largest cohort ever reported to deeply characterize phenogenotypically *PLA2G6*-related parkinsonism. This study performed a systematic literature review and enrolled 86 cases from the world, including 16 patients from Europe [8]. Thus, the patients with *PLA2G6*-related parkinsonism from Europe were referenced from the review of Magrinelli et al. [8]. Information not available in the patient files was noted as missing. The genotypes and phenotypes of these patients were summarized. Statistical analyses were performed with R version 3.5.3 software. Differences in the distribution of clinical features between the Chinese and European groups were assessed. Continuous data were compared across cohorts using an unpaired *t* test unless the data were not normally distributed (as assessed by the Kolmogorov–Smirnov test). In these instances, the Mann–Whitney U test was used. Categorical data were compared using the Chi square test or Fisher exact test wherever appropriate. A value of *p* < 0.05 (two-sided) was considered statistically significant.

## 3. Results

### 3.1. Clinical Features of the Patients

All six probands were born at term following an uneventful pregnancy and had normal early developmental milestones. The charts of the PLAN families are separately shown in Figure 1A. Consanguineous marriage did not occur in any of these families. The detailed clinical characteristics for these six probands are summarized in Table 1. Among them, four index patients were males and two were females. The mean age of disease onset was 22.7 years (6–35 years), and five had onsets with abnormal gait. The youngest patient (Case 1) mimicked a phenotype of HSP, whereas the other five cases had a phenotype mimicking parkinsonism. Brain MRI demonstrated that all patients had cerebellar atrophy and three had iron accumulation on the globus pallidus bilaterally and the substantia nigra.

Case 1 (Family 1 II-1, Figure 1A) was a 13-year-old girl presenting with a 7-year history of abnormal walking posture. In addition, she frequently fell down and had a poor academic performance in school. Neurological examinations revealed spastic gait, pes cavus, increased muscle tension in the lower limbs, and a positive Babinski sign. Brain MRI revealed atrophy of the cerebellum (Figure 2A1), T2 weight, and hypo-intensity in the globus pallidus bilaterally and in the substantia nigra (Figure 2A2,A3). SWI (susceptibility weighted imaging) showed evidence of increased iron in the substantia nigra and globus pallidus (Figure 2A4,A5). At the last follow-up, the patient was 16 years old, and her condition had not worsened after exercise and medication (baclofen) adherence.

Case 2 (Family 2 II-3, Figure 1A) was a 25-year-old male with gait instability for 10 years. Initially, he noticed only mild walking instability and occasional dizziness. There was no muscle weakness and no tremor in his limbs. At the age of 23, he developed slurred speech and the symptoms of gait disturbance were more evident, including easy falls when turning quickly and difficulty running or climbing stairs. In relation to study at school, he also exhibited mild learning difficulties, especially in mathematics, and dropped out after grade 7. Moreover, he grew prone to skin allergies with recurring episodes of skin rash all over the body. Physical examinations revealed horizontal nystagmus, dysarthria, and absent tendon reflexes. In addition, he presented with a thin stature (weight 45 kg and height 169 cm), microcephalic (head circumference 52 cm; normal 54–58 cm), special face (high eyebrow arches, deep eye sockets, abnormal tooth arrangement, and protruding jaw; Appendix A), skin hyperpigmentation and rash over the whole body (Appendix A), and scoliosis (Appendix A). His cognition was impaired with a Mental State Examination (MMSE) score of 19/30. The patient was negative for anti-O antibody, rheumatoid factor, and anti-neutrophil cytoplasmic antibody. Brain MRI revealed atrophy of the cerebellum (Figure 2B1) and iron deposition in the globus pallidus bilaterally (Figure 2B2,B4) and substantia nigra (Figure 2B3,B5). No significant abnormalities were recorded in the EMG.

Case 3 (Family 3 II-2, Figure 1A) was a 42-year-old male who had progressive walking difficulty for 22 years. He was a printing factory worker without exposure to any toxins. At the age of 20, he began to experience walking instability on uneven surfaces, mild rigidity of the lower limbs and right lower extremity weakness. The symptoms were progressively exacerbated with age, and he required a walking aid at 39 years old. He gradually developed dysarthria, dysphagia, and difficulties with mouth opening at 40 years old. Neurological examinations showed a mask-like face, unresponsiveness, decreased muscle strength and increased muscle tone of the limbs, absent tendon reflexes in the lower extremities, decreased vibration and proprioception, and positive ataxia sign. The MoCA score was 1, and the MMSE score was 4 (primary school degree). Brain MRI revealed apparent atrophy of cerebellum (Figure 2C1), but no obvious iron deposition was observed (Figure 2C2–C5). EMG demonstrated that bilateral lower extremities affected peripheral sensorimotor polyneuropathy with predominant distal motor demyelinating neuropathy. The patient was prescribed levodopa which resulted in improvement in speech and motor symptoms. His 44-year-old brother had only mild walking instability in a straight line and mild intelligence impairment, however, these features did not affect his life or work.

Case 4 (Family 4 II-1, Figure 1A) was a 29-year-old male who initially showed slow and unstable walking with weakness of the lower limbs at the age of 28 years. He developed parkinsonism, including slurred speech, bradykinesia, tremor, stiffness, and difficulty walking or standing, at 30 years of age. Moreover, cognitive and neuropsychiatric decline (depressed mood and decreased appetite) was also observed. The Unified Parkinson’s Disease Rating Scale (UPDRS) scores of the patients before and after levodopa treatment were 29 and 23, respectively. Physical examinations showed a mask-like face, tremor of the left hand, increased muscle tone and tendon reflex of the limbs, and positive Babinski’s and ataxia signs. Brain MRI showed atrophy of the cerebellar and pons medulla, and no abnormal signs of SWI (Figure 2D1–D5). 

Case 5 (Family 5 II-1, Figure 1A) was a 32-year-old male who complained of a one-year history of slurred speech and walking instability accompanied by occasional dizziness and tremor of the left hand. The symptoms were progressively aggravated. Levodopa was prescribed, and his walking instability was slightly improved. Physical examinations showed a MoCA score of 18, mask-like face, slightly limited upper vision of both eyes, increased muscle tension and tendon reflex of upper limbs, decreased tendon reflex of lower limbs, negative Babinski’s signs, and positive ataxia signs. Brain MRI suggested brain atrophy, but SWI showed no abnormalities (Figure 2E1–E5). 

Case 6 (Family 6 II-1, Figure 1A) was a 40-year-old female with progressive instability who slowly walked for 5 years. Obvious swaying, mild lethargy, and memory decline were noted at the first visit. Examinations revealed obvious hyper-reflexia of the tendon and positive Babinski’s and ataxia signs. Brain MRI showed cerebellar atrophy (Figure 2F1) and no obvious iron deposition (Figure 2F2–F5). After 3 years of follow up, she developed difficulties with mouth opening, dysarthria, decreased muscle strength with superimposed severe dystonic movements and contractures of both hands, and became wheelchair bound.

### 3.2. Identification of Variants by WES

Four novel variants, c.967G>A (p.V323M), c.1450G>T (p.D484Y), c.1631T>C (p.M544T), and c.1915delG (p.A639Qfs*27) (Figure 1B), and five previously reported pathogenic mutations in *PLA2G6*, c.991G>T (p.D331Y), c.1077G>A (p.M358IfsX), c.1117G>A (p.G373R), c.1427+2T>A, and c.1670C>T (p.S557L) [6,18,19,20], were identified in six PLAN index cases. All of these patients harbored compound heterozygous variants that co-segregated with the disease (Figure 1A). Four novel variants were absent in 1000 G, ExAC, and 2022 in-house controls, and showed high conservation between different species (Figure 1B). The missense novel variants were predicted to be deleterious by SIFT, PolyPhen-2, and Mutation Taster. According to ACMG standards, c.1915delG was classified as a pathogenic variant, and c.967G>A, c.1450G>T, and c.1631T>C were classified as likely pathogenic variants (Appendix A). All six cases had no other pathogenic or likely pathogenic mutations in genes known to be associated with early-onset atypical movement disorder. 

### 3.3. Founder Effect of the c.991G>T Mutation among Chinese Patients

On haplotype analysis, five patients (including a patient previously reported by our center) were compound heterozygotes for the c.991G>T allele with other mutant *PLA2G6* alleles (Table 2). Twenty-three SNPs (listed in the leftmost column of Table 2) located upstream and downstream of the mutation were selected to construct haplotypes for testing the gene identity by descent, and two generations of five families were tested. According to the results of SNP testing, the size of the shared haplotype was at least 717 kb (chromosome 22: 37,905,632–38,622,598) (Table 2). All five alleles containing the c.991G>T mutation shared a common haplotype surrounding it, suggesting the presence of a founder effect.

### 3.4. Genotypes and Phenotypes of Chinese PLAN Patients

Based on a retrospective analysis, 69 PLAN patients from 65 Chinese families were included in the present study, and these patients carried 47 different *PLA2G6* mutations (Appendix A). Combining six patients with nine mutations in our study, we found that the most frequent mutations in *PLA2G6* were c.991G>T (p.D331Y, 26.67%), c.1077G>A (p.M358IfsX, 6.67%), c.1117G>A (p.G373R, 5.33%), c.1427+1G>A (5.33%), and c.1A>G (p.M1V, 5.33%) (Appendix A). The most frequent phenotypes were INAD, *PLA2G6*-related parkinsonism, and ANAD, accounting for 44%, 41.33%, and 5.33% of cases (total 75), respectively. Overall, 22 patients in the Chinese population contributed to the phenogenotypic description of *PLA2G6*-related parkinsonism (Appendix A). The most frequent clinical manifestations were parkinsonism (100%), bradykinesia (95.45%), rigidity (81.82%), gait disturbance (77.27%), psychiatric features (54.55%), rest tremor (50%), dystonia (50%), and postural impairment (50%). When further compared with the European population (16 cases) (Appendix A), the frequency of psychiatric features, cognitive decline, and myoclonus were significantly greater than that of the Chinese population (Appendix A).

## 4. Discussion

PLAN is often misdiagnosed due to complicated clinical manifestations, and few studies have been conducted to investigate the genetic and clinical features of PLAN in Chinese patients due to the low incidence. In this study, nine mutations, including four novel mutations, were identified in six unrelated Chinese PLAN families. The potential underlying founder effect was verified for the first time in the Chinese population. Simultaneously, we summarized the genotypic and phenotypic profiles of PLAN patients in China.

Here, we made the genetic diagnosis in one patient with a phenotype of HSP and five with parkinsonism disease and some atypical clinical characteristics. Case 1 presented with spasticity, hyperreflexia of the lower limbs, and bilateral extensor plantar response, which suggested HSP. Growing evidence suggests that PLAN is a phenotypic continuum. Since the three major phenotypes had overlapping clinical and radiological features [6], we hypothesize that the symptoms that initially present as HSP may represent as “pre-parkinsonism” in PLAN patients. In addition to the nervous system, case 2 developed special facial features, skeletal deformities, microcephaly, and systemic dermatitis, which have been rarely reported in *PLA2G6*-related parkinsonism patients to date. However, the three former features are noted in 21.4%, 35.7%, 17.9% of INAD patients, respectively [9]. Even though immune disorders are rare among patients with PLAN, the protein encoded by *PLA2G6* plays a role in inflammation and immune responses [1], and *PLA2G6* knockout mice also show evidence of neuroinflammation [21,22]. Therefore, immune function problems may be underestimated in the clinic. If this condition is confirmed among more patients in the future, this information will be very helpful in guiding the research for the development of new therapeutic options. Intriguingly, in family 3, the proband exhibited an earlier onset and developed a more severe course than his older brother. It was also reported previously that even patients from one family with the same genotype may have differences in phenotype [23]. Therefore, in addition to protein dysfunction, other environmental or unknown genetic factors may modulate these phenotypes. Based on this study, levodopa can be effective in *PLA2G6*-related parkinsonism patients for some periods of time, but further clinical observations are required to determine its long-term efficacy. 

Pathologically, PLAN is characterized by the depletion of cerebellar cortical neurons accompanied by astrocytosis, axonal spheroids in the central and peripheral nervous system, and progressive brain iron deposition [2,8]. In our study, only half of the patients (3/6) had brain iron deposition, but all of them (6/6) had cerebellar atrophy. Cerebellar atrophy is often the earliest sign on MRI, whereas evidence of brain iron deposition in the basal ganglia appears later [2,9]. In the dystonia-parkinsonism phenotype, brain iron accumulation was reported in one-third of patients [10]. This view was also supported by our results. 

The *PLA2G6* gene encodes a group of VIA calcium-independent phospholipase A2 (iPLA2β) enzymes that are involved in a variety of cellular functions, including phospholipid metabolism, membrane homeostasis, calcium signal transduction, mitochondrial function, apoptosis, and cell proliferation [1,24]. The amino acid sequence of iPLA2β (Transcript NP_003551.2) codes for diverse functional domains including seven N-terminal ankyrin repeat domains (amino acids: 150–382), a highly conserved patatin-like phospholipase domain (amino acids: 480–793) that harbors a nucleotide-binding domain (centered at amino acid 485), a serine lipase consensus sequence (GXSXG) (S519), and the C-terminal calmodulin-binding region (amino acids 747–759) [25,26]. Of the four novel variants identified in this study, p.V323M was located in the ankyrin repeats domain, and p.D484Y, p.M544T, and p.A639Qfs*27 were located in the patatin-like phospholipase domain, and may therefore impair the function of iPLA2β through multiple loss-of-function mechanisms, affecting either its enzyme activity, regulation, or interactions at the macromolecular level. However, the specific mechanism still needs further study.

In this study, the c.991G>T mutation in *PLA2G6* commonly observed in the Chinese population [6,12] was analyzed by SNP-based haplotype analysis to estimate whether this mutation was due to a founder effect. The five informative Chinese families with the *PLA2G6* c.991G>T mutation shared an identical TSNP haplotype covering a chromosomal region of approximately 717 kb. All five known Chinese families are Han populations from South China and these families are not related based on available genealogical information at that time. The shared haplotype indicates that the *PLA2G6* c.991G>T mutation frequently observed in the Chinese population seems to be a founder mutation. However, future research with more patients is required to determine this further. The identification of common founder mutations can greatly facilitate the molecular diagnosis of PLAN, which allows targeted mutational analysis of specific gene regions as the first step in a genetic testing strategy. 

The clinical characteristics of *PLA2G6*-related parkinsonism were different between the Chinese and European populations. Three reasons may need to be considered. First, in the Chinese patients with PLAN, myoclonus, cognitive decline, and psychiatric features may be masked by complex phenotypic features. This highlights the need to evaluate this aspect in future research. Second, PLAN is a rare disease, and most studies in the Chinese population have relatively small sample sizes. Multicenter studies with large samples may help to clearly identify the characteristics of *PLA2G6*-related parkinsonism in the population. Finally, different genetic backgrounds may be the main reasons for the discrepancy between Chinese and European populations, such as the different founder effects identified in this and previous studies. 

## 5. Conclusions

In conclusion, we identified four novel and five previously reported mutations in six PLAN patients, and documented the typical and atypical clinical characteristics of these patients. We further demonstrated that c.991G>T, the most common *PLA2G6* mutation in China, may represent a rare disease-associated founder mutation. This study expands the clinical and genetic spectrum of PLAN disease and provides an insightful view of the founder effect, which might have considerable diagnostic and therapeutic implications in the near future.

## Figures and Tables

**Figure 1 brainsci-12-00517-f001:**
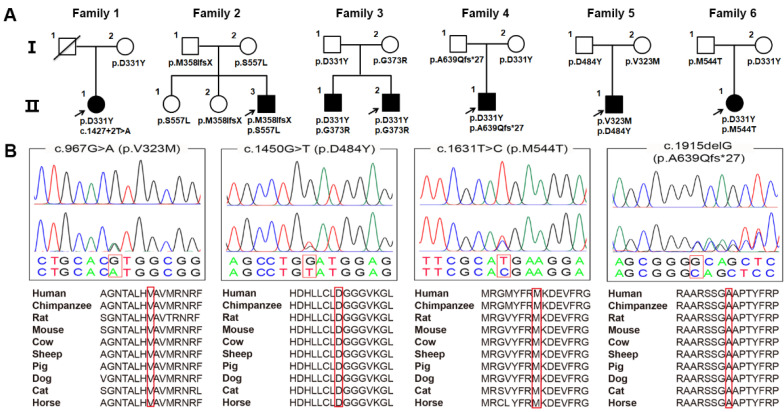
Pedigree chart, chromatogram and homology comparison. (**A**) Pedigree chart of six PLAN families. The arrows indicate the proband, and the diagonal lines indicate deceased members. Squares indicate males, and circles indicate females. The black squares or circles indicate affected individuals. (**B**) Chromatogram and homology comparison of novel mutations in *PLA2G6*. The upper chromatograms represent the reference sequence and the lower chromatograms represent the mutated sequence. Highlighted zones represent the four novel mutations among eight species.

**Figure 2 brainsci-12-00517-f002:**
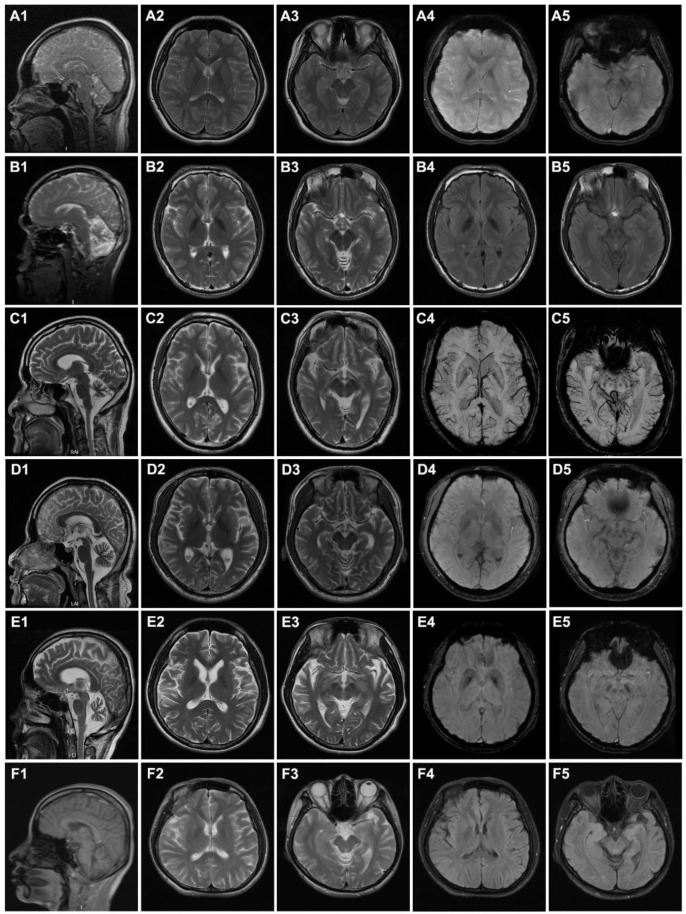
Brain MRI examination of 6 probands. (**A**–**F**) indicate cases 1–6, respectively. (**A1**–**F1**) Sagittal images demonstrate cerebellar atrophy in all patients; (**A2**–**F2**,**A3**–**F3**) T2-weighted images show the globus pallidus and substantia nigra; (**A4**,**A5**,**C4**–**E4**,**C5**–**E5**) The SWI (susceptibility weighted imaging) image shows the globus pallidus and substantia nigra; (**B4**,**B5**,**F4**,**F5**) T2-Flair images show the globus pallidus and substantia nigra.

**Table 1 brainsci-12-00517-t001:** Detailed clinical features of six probands with PLAN.

Patient	Case 1	Case 2	Case 3	Case 4	Case 5	Case 6
Gender	F	M	M	M	M	F
Age at onset (year)	6	15	20	29	31	35
Disease duration (years)	7	10	22	1	1	5
Family History	No	No	Yes	No	No	No
Province of origin	Jiangxi	Anhui	Zhejiang	Zhejiang	Anhui	Zhejiang
Variants	p.D331Y, c.1427+2T>A	p.M358IfsX, p.S557L	p.D331Y, p.G373R	p.D331Y, p.A639Qfs*27	p.V323M, p.D484Y	p.D331Y, p.M544T
Initial symptoms	Gait disturbance	Gait disturbance	Slowly walking	Gait disturbance	Slurred speech	Slowly walking
Abnormal posture and gait	+	+	+	+	+	+
Difficulty walking	+	+	+	+	−	+
Parkinsonism	−	+	+	+	+	+
Bradykinesia	−	+	+	+	−	+
Rest tremor	−	−	−	+	+	+
Rigidity	−	+	+	+	+	+
Dysarthria	−	+	+	+	+	+
Cognitive decline	+	+	+	−	+	+
Psychiatric symptoms	−	−	−	+	−	+
Autonomic dysfunction	−	+	−	+	+	+
Sensory dysfunction	−	+	+	+	−	−
Nystagmus	−	+	−	−	−	−
Eye movement abnormality	−	+	+	−	+	−
Dysphagia	−	−	+	−	−	+
Dystonia	+	+	+	+	+	+
Myoclonus	−	−	−	−	−	−
Muscle strength decline	−	+	+	−	−	+
UL/LL Tendon reflexes	++/++	−/−	++/+++	+++/++	+++/+	+++/+++
Babinski’s sign	+	−	+	+	−	+
Ataxia signs	−	+	+	−	+	−
Cerebellar atrophy	+	+	+	+	+	+
Cerebral atrophy	−	−	+	−	+	+
Iron deposition in globus pallidus	+	+	−	−	+	−
Benefit from Levodopa	NA	NA	+	+	+	NA

Abbreviations: PLAN: *PLA2G6*-associated neurodegeneration; F: female; M: male; +: present/abnormal; −: absent; UL: upper limb; LL: lower limb; NA: not available.

**Table 2 brainsci-12-00517-t002:** SNP-based genotype and haplotypes of the five families with *PLA2G6* c.991G>T.

TSNP	MAF	Family 1				Family 3						Family 4						Family 6						Previous Report Family					
		Case 1		Mother		Case 3		Mother		Father		Case 4		Mother		Father		Case 6		Mother		Father		Case		Mother		Father	
22-37462926-G-A	0.429	G	G	G	G	G	A	G	A	G	A	G	A	G	A	G	A	A	A	A	A	A	A	G	G	G	G	G	G
22-37528576-A-G	0.467	A	A	A	A	A	A	A	G	A	G	A	G	G	G	A	A	G	A	G	A	G	A	A	A	A	G	A	G
22-37603390-C-T	0.324	C	T	T	T	C	T	C	T	T	T	C	T	C	T	C	C	C	T	C	T	C	T	T	T	T	T	C	T
22-37707962-C-T	0.424	C	T	C	T	T	C	C	C	T	T	C	T	C	T	T	T	T	T	T	T	T	T	C	T	C	T	C	T
22-37825220-G-C	0.4	G	C	G	C	G	C	G	C	G	C	G	G	G	G	G	C	C	C	C	C	C	C	G	C	C	C	G	C
22-37905632-G-A	0.495	G	G	A	A	G	G	G	G	G	G	G	G	G	G	G	G	G	G	G	G	G	G	A	G	G	G	A	A
22-38032709-A-G	0.49	G	A	G	A	A	A	A	G	A	A	G	A	G	A	G	G	G	A	G	A	G	A	G	A	A	A	G	G
22-38122448-C-T	0.381	C	T	T	T	T	T	C	T	C	T	T	T	T	T	T	T	C	T	C	T	C	T	T	T	T	T	T	T
22-38237562-C-A	0.486	A	A	A	A	A	A	A	A	A	A	A	A	A	A	A	A	A	A	A	A	A	C	A	A	A	A	A	A
22-38303155-C-T	0.486	C	T	C	T	T	T	C	T	C	T	T	T	T	T	T	T	C	T	C	T	C	T	C	T	T	T	C	C
22-38463968-A-G	0.305	G	G	A	G	G	G	A	G	G	G	G	G	G	G	G	G	A	G	A	G	A	G	G	G	G	G	G	G
22-38528924-G-T		G	T	G	T	G	T	G	G	G	T	G	T	G	T	G	G	G	T	G	T	G	G	G	T	G	T	G	G
22-38543453-T-C	0.357	C	T	T	T	C	T	T	T	T	T	T	T	C	T	T	T	T	T	T	T	T	T	T	T	T	C	T	C
22-38622598-C-A	0.357	A	C	C	C	C	C	C	A	A	C	C	C	C	C	C	C	C	C	A	C	C	C	C	C	C	A	C	A
22-38723050-C-T	0.376	C	T	T	T	C	C	C	C	C	C	C	C	C	C	C	T	T	C	T	C	C	C	C	C	C	C	C	C
22-38829774-G-C	0.367	G	G	G	C	G	C	G	C	G	C	C	C	G	C	G	C	G	C	G	C	G	G	G	G	G	G	G	G
22-38918894-T-G	0.39	T	T	T	T	G	T	T	G	G	G	G	G	T	G	T	G	G	T	T	T	T	G	T	G	T	G	T	T
22-39029695-T-C	0.486	T	T	C	T	T	T	T	T	T	T	T	T	T	T	T	C	C	T	T	C	T	C	C	C	T	C	T	C
22-39157384-G-A	0.467	A	G	G	G	A	A	A	A	A	A	A	G	G	G	A	A	A	G	G	G	G	A	G	G	G	A	G	A
22-39258795-G-A	0.457	C	G	G	G	G	G	G	G	G	G	G	G	G	G	G	G	C	G	G	G	G	C	C	G	G	C	C	C
22-39355415-A-G	0.500	G	A	G	A	G	G	G	G	G	G	A	G	G	G	G	A	A	A	A	A	A	A	A	A	G	A	G	A
22-39449014-C-T	0.429	C	C	C	C	C	C	C	C	C	C	C	C	C	C	C	C	C	C	C	C	C	C	C	C	C	C	C	C
22-39551989-G-A	0.467	G	G	G	G	G	G	G	G	G	G	G	G	A	G	A	A	A	G	G	G	G	A	G	G	G	G	G	A
22-39663185-A-G	0.386	G	G	G	G	G	G	G	G	G	G	G	G	A	G	A	G	G	G	A	G	A	G	G	G	G	G	G	G

Abbreviations: SNP: Single nucleotide polymorphism; TSNP: Tag single nucleotide polymorphism; MAF: minor allele frequency; Marked with yellow and blue shading-Haplotypes of PLA2G6 c.991G>T (indicated in red type-face).

## Data Availability

The data presented in this study are available on request from the corresponding authors.

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
