# Peer review of "Clinical Characterization and Founder Effect Analysis in Chinese Patients with Phospholipase A2-Associated Neurodegeneration"

_brainsci, 2022, doi:10.3390/brainsci12050517_

Round 1

Reviewer 1 Report

The manuscript presents the detailed characteristics of 5 cases with rare PLAN disorder. It is valuable part of the article describing patients with different subtypes of the disease. Moreover, novel pathogenic PLA2G6 variants are identified. However, I would like to mention the minor concerns:
In the Material and Methods section there is a no information about the lack of consanguinity among probands (it appears later on). 

The number of 5 patients is too small to unambigously conclude that this a common founder effect in Chinese population.

In Table 2 the case 1 has a haplotype with G, but his mother bearing the same mutation has A in a position 22 - 37905632 -G-A. 

Besides, the manuscript presents a clinical and scientific value in the field of neurodegenerative disorders.

Reviewer 2 Report

The authors present a nice, thorough study examining how genetics can  contribute to eteiology of PLAN. The text is well written, the figures are nicely presented and the conclusions are sound. 

Comments:

1) Given the use of WES, did the authors exclude pathogenic mutations in other known genes that are have been implicated in early-onset atypical movement disorder? If so, this needs to be stated to rule out other contributing factors.  

2) Have the authors considered whether the various mutations described in their manuscript will impact the expression/stability of  Pla2g6? This would be informative to the reader as several different pathogenic mutations are described within the article. 

On line 265, the sentence "There are only 50% patients has brain iron
deposition, but 100% has cerebellar atrophy in our study" is somewhat confusing and could be improved. 

The discussion could be improved slightly by describing some of the known biology associated with PLA2G6, and which pathways the aforementioned mutations might disrupt. 

Reviewer 3 Report

Hao-Ling Cheng and co-authors report six Chinese families (7 affected subjects) with PLA2G6-associated neurodegeneration describing four novel likely pathogenic PLA2G6 variants. In addition, they performed a haplotype analysis of the c.991G>T D331Y PLA2G6 variant studying 23 tagging SNPs and found a shared haplotype confirming a founder effect in the Chinese population. The manuscript is clear and interesting. However, I have several major and minor remarks.

Major remarks:

The section comparing the European and Chinese PLAN patients needs to be improved.

- It is not clear to the reviewer how the 22 Chinese and 16 European PLAN patients were selected for the comparison. The age difference of the two cohorts was significantly different? Please clarify in the methods section.

- Myoclonus and psychiatric features can be under-reported features in complex neurological cases. Are these clinical features explicitly excluded in the Chinese cohort?  

- In the table S2 cognitive decline seems to be more prevalent in Europeans than in Chinese PLAN patients. Why was it not included in the manuscript?

- The reviewer cannot find in the results the correlation analyses between PLA2G6 variants and PLAN phenotypes as stated the methods section.

Minor remarks:

- English language revision by a native speaker is needed.

- I re-checked the ACMG classification of the novel variants (tool: VarSome.com):

PLA2G6(NM_003560.4): c.967G>A (p.Val323Met) = VUS (PM2, PP2)

PLA2G6(NM_003560.4): c.1450G>T (p.Asp484Tyr) = Pathogenic (PM1, PM2, PM5, PP2, PP3)

PLA2G6(NM_003560.4): c.1631T>C (p.Met544Thr) = Pathogenic (PM3, PM1, PM2, PP2, PP3)

PLA2G6(NM_003560.4): c.1915del (p.Ala639GlnfsTer27) = Pathogenic (PM3, PVS1, PM2)

Round 2

Reviewer 3 Report

I thank the authors for their efforts to improve their manuscript. I have no additional remarks.